# Targeting of the Peritumoral Adipose Tissue Microenvironment as an Innovative Antitumor Therapeutic Strategy

**DOI:** 10.3390/biom12050702

**Published:** 2022-05-14

**Authors:** Melania Lo Iacono, Chiara Modica, Gaetana Porcelli, Ornella Roberta Brancato, Giampaolo Muratore, Paola Bianca, Miriam Gaggianesi, Alice Turdo, Veronica Veschi, Matilde Todaro, Simone Di Franco, Giorgio Stassi

**Affiliations:** 1Department of Health Promotion Sciences, Internal Medicine and Medical Specialties (PROMISE), University of Palermo, 90127 Palermo, Italy; melania.loiacono@unipa.it (M.L.I.); gaetana.porcelli@unipa.it (G.P.); paola.bianca@unipa.it (P.B.); alice.turdo@unipa.it (A.T.); matilde.todaro@unipa.it (M.T.); 2Department of Surgical Oncological and Stomatological Sciences (DICHIRONS), University of Palermo, 90127 Palermo, Italy; chiara.modica@unipa.it (C.M.); ornirobert.brancato@gmail.com (O.R.B.); muratore96@gmail.com (G.M.); miriam.gaggianesi@unipa.it (M.G.); veronica.veschi@unipa.it (V.V.)

**Keywords:** tumor microenvironment, adipose tissue, cancer stem cells, adipokines, exosomes, target therapy

## Abstract

The tumor microenvironment (TME) plays a key role in promoting and sustaining cancer growth. Adipose tissue (AT), due to its anatomical distribution, is a prevalent component of TME, and contributes to cancer development and progression. Cancer-associated adipocytes (CAAs), reprogrammed by cancer stem cells (CSCs), drive cancer progression by releasing metabolites and inflammatory adipokines. In this review, we highlight the mechanisms underlying the bidirectional crosstalk among CAAs, CSCs, and stromal cells. Moreover, we focus on the recent advances in the therapeutic targeting of adipocyte-released factors as an innovative strategy to counteract cancer progression.

## 1. Introduction

In the last decade, considerable interest has focused on the contribution of the tumor microenvironment (TME) to cancer progression and chemotherapy resistance. The malignant phenotype is not exclusively driven by a specific cancer cell subset but rather regulated by a complex interplay between cancer stem cells (CSCs) and TME elements [1]. This bidirectional crosstalk includes signals that promote tumor growth, foster metastasis dissemination, and limit drug penetration and uptake. In this context, the scientific community’s interest in the role of adipose tissue (AT) in carcinogenesis is constantly increasing. Indeed, it is known that AT is involved in the deregulation of inflammatory and immune responses, leading to aberrant metabolism, and contributing to cancer development and progression [2]. The role of AT in cancer progression is supported by its anatomical distribution, such as in breast cancer, where the TME predominantly comprises adipose tissue, with cancer cells growing in contact with adipose cells [3]. The tight and prolonged contact between cancer and adipose cells leads to the reprogramming of adipocytes, with the generation of the so-called ‘cancer-associated adipocytes’ (CAAs), which can directly or indirectly facilitate tumor growth and progression by releasing adipokines, growth factors, and metabolites. Understanding the crosstalk between CAAs and CSCs, and how adipokines and metabolites released in TME can regulate different molecular pathways may be essential in the prevention of tumor progression. Here, we report the role of AT in promoting cancer growth and describe the different adipokines and metabolites involved in the interplay between CAAs and CSCs. Moreover, we highlight the interaction between immune cells and CSCs, and putative candidates/tools for therapeutic interventions.

## 2. Cellular and Functional Heterogeneity of Adipose Tissue

Since the discovery of leptin by Zhang et al. in 1994 [4], AT has been considered the biggest human endocrine organ. AT is characterized by high cellular plasticity, which is finely regulated during the different stages of embryonic human development. AT regulates different physiological processes, such as hormonal balance, inflammation, hypoxia, and metabolism, and also plays a crucial role in tumor initiation, promotion, and progression [5]. AT can be classified into white (WAT) and brown adipose tissue (BAT), with each type being endowed with different grades of cellular heterogeneity and functions [6,7].

WAT is mainly composed of unilocular adipocytes that contain a singular big lipid droplet, and is characterized by rich vascularization consisting of capillaries that divide the tissue into multiple lobules. WAT is localized at both the subcutaneous (SAT) and visceral levels (VAT), with a distribution that changes at different ages and according to sex [8]. Although the principal functions of WAT include lipid storage and mechanical support of inner organs, WAT is considered an active organ that has endocrine functions due to the production of adipokines, which include hormones (leptin, adiponectin) and cytokines, such as interleukin-6 (IL-6), IL-1β, IL-5, IL-8, MCP1 (monocyte-chemoattractant protein-1), tumor necrosis factor-α (TNF-α), interferon-ү (INF-γ), hepatic growth factor (HGF), plasminogen activator inhibitor-1 (PAI-1), and c-c motif chemokine ligand-2 (CCL2). All these released factors play a crucial role in cancer initiation and progression, thus regulating the inflammatory process and angiogenesis [9,10].

On the other hand, BAT is mainly composed of brown multilocular adipocytes, characterized by numerous small lipid droplets and mitochondria. It has been demonstrated by positron-emission tomography (PET) and tissue biopsy that BAT is localized to the supraclavicular and parathyroid areas [11,12]. Cinti et al. demonstrated that within rats’ BAT, two different subsets of brown adipocytes can be identified, endowed with different uncoupling protein 1 (UCP1) expression levels and thermogenic functions, which is particularly evident following exposure to acute and chronic cold [13]. Recently, Song and colleagues showed that two different subsets of brown adipocytes coexist in murine adipose tissue, characterized by low and high thermogenic function. Low-thermogenic brown adipocytes express lower levels of UCP1 and ADIPOQ (adiponectin, C1Q, and collagen domain containing), contain larger lipid droplets, and have a lower number of mitochondria [14]. A further level of complexity regarding the cellular heterogeneity is given by the presence, within BAT, of adipocytes showing an intermediate phenotype, defined as beige or brown in white (brite). Although sharing their thermogenic functions, brown and beige adipocytes are considered distinct cell types due to their different developmental origins and anatomical locations [15]. BAT, through autocrine, paracrine, and endocrine mechanisms, plays a role that is not exclusive to the body’s thermogenic regulation, as it is also implicated in glucose uptake and lipid metabolism [16,17]. Moreover, BAT releases a plethora of cytokines called ‘batokines’, including insulin growth factor-1 (IGF-1), fibroblast growth factor-21 (FGF-21), IL-6, and endothelial growth factor-A (EGF-A), which play a role in tissue remodeling and thermogenic function [18,19].

Both WAT and BAT mature adipocytes are derived from adipose-derived mesenchymal stem cells (AD-MSCs). AD-MSCs are able to self-renew and differentiate into the mesodermal cellular lineages, such as neurons, myocytes, chondrocytes, osteoblasts, and adipocytes [20]. SAT and VAT adipocytes have different origins, with VAT cells deriving from the mesothelium and identified by the expression of WT1, unlike SAT and BAT cells [21].

It has recently been reported that a specific excess of visceral fat in cancer patients is correlated with a worse prognosis and predicts a lower response to antitumor treatments [22]. Therefore, different degrees of correlation between tumor risk and progression and different types/amounts of fat exist.

## 3. Obesity and Cancer: The Cause–Effect Relationship

The clinical impact of obesity is still controversial, without clear evidence of its role in cancer progression. Recently, several research groups have investigated the molecular mechanisms underlying the crosstalk between adipocytes and cancer cells.

Obesity is a complex and chronic disease, and is considered as one of the most common diseases in the Western world. It is currently recognized as a global pandemic. According to recently published data by the World Health Organization (WHO), in the last 40 years, a 3-fold increase in obesity in adults has been observed in industrialized countries, and a further increase is expected in the near future [23].

Several studies supported by the International Agency for Research on Cancer (IARC) established that the highest BMI (≥30 kg/m^2^) is correlated with an increased risk of developing different types of solid tumors, including esophageal, meningioma, prostate, ovarian, gallbladder, breast, colorectal, kidney, thyroid, and pancreatic ductal cancer; reduced overall survival; and increased risk of recurrence [24,25].

Obese post-menopausal women show a 30% higher risk of developing ER-positive breast cancer compared to normal-weight subjects [26,27], and obesity in men is considered the major risk factor for colon cancer development [28]. In this regard, in a large population-based cohort study, Bhaskaran and colleagues observed a decrease in the overall survival in colorectal cancer (CRC) patients with BMI between 25 and 50 kg/m^2^ when compared to CRC patients with a lower BMI [29]. Although the obesity–cancer relationship has been widely shared by the scientific community, the molecular mechanisms underlying this biological/clinical phenomenon are still not clear [30].

One of the possible obesity-driven mechanisms supporting cancer disease concerns the onset of a chronic inflammatory state and increased cellular oxidative stress. This biological process affects the genome stability of cells in a direct and indirect way by driving the accumulation of mutations due to the increased presence of oxygen reactive species (ROS), and increasing lysine homocysteinylation in DNA damage repair systems’ proteins through post-translational modification [31]. The enhanced induction of unrepaired mutations plays a crucial role in both the initiation and progression of the tumor by contributing to cancer cells’ heterogeneity, with a direct impact on cell survival [32]. However, some studies have reported a better prognosis in obese patients with premenopausal breast cancer, non-small cell lung cancer (NSCLC), and head/neck cancers, a phenomenon called ‘the obesity paradox’, which is likely driven by the inadequacy/inconsistency of data collection/analysis, and the selection and stratification of patients [33,34].

It has been highlighted that there is continuous and dynamic crosstalk between adipose and tumor cells, which is mainly sustained by the altered production of steroid hormones, adipokines, and cytokines, which affects cancer cells during all steps of tumor progression [35,36,37]. Among the different types of adipokines, leptin and adiponectin are considered the main regulators of the dynamic control of appetite and energy expenditure [38,39,40]. In obesity, higher leptin levels and reduced adiponectin levels were observed compared to normal weight individuals. Moreover, WAT secretes a plethora of proinflammatory and protumorigenic factors that recruit different types of immune and stromal cells, thus creating a tumor-permissive microenvironment [41,42,43,44]. In the context of the adipose tissue of breast cancer patients, it has been observed that macrophages recalled by inflammation phenomena (such as pyroptosis and necrosis of adipocytes) can form crown-like structures (CLSs), which is highly correlated with relapse and mortality [45,46].

Recently, we demonstrated that visceral AT stromal cells (V-ASCs), through the release of IL-6 and HGF, induce transcriptional reprogramming of CRC cells toward the acquisition of a mesenchymal-like phenotype [47]. Moreover, V-ASCs’ released factors lead to an increase in CD44v6^+^ cells, which in turn attract ASCs within the tumor bulk due to their release of neurotrophins. The recruited ASCs, following the release of VEGF by CD44v6^+^ cancer cells, actively participate in tumor progression processes by transdifferentiating into endothelial-like cells [47].

The clinical impact of obesity is still controversial, without clear evidence of its role in cancer progression. Obese cancer patients are refractory to standard chemotherapy. This phenomenon could be driven by a direct effect of adipose tissue on cancer cell behavior, and the failure of clinicians to provide adequate treatment doses due to possible toxicity [48]. In this regard, the ASCO 2012 guidelines recommend the patient’s current weight as the best method for assessing the amount of chemotherapy to prescribe. However, in obese cancer patients, given the high amount of treatment provided, pharmacodynamics and pharmacokinetics studies should be performed to increase the efficacy of therapeutic options, and limit associated side effects [48,49,50]. In 2021, new updates about the treatment of obese adult patients were provided by ASCO, in which immunotherapies and new targeted anticancer therapies were included [51].

## 4. Cancer-Associated Adipocytes (CAAs): Origin and Role in Cancer Progression

Numerous pieces of evidence support the active contribution of TME, with its cellular and acellular elements, to cancer cell proliferation, invasion, epithelial to mesenchymal transition (EMT), and response to therapies. TME includes a wide variety of cell types, such as endothelial cells, fibroblasts, pericytes, immune cells, mesenchymal stromal cells (MSCs), and mature adipocytes [52,53].

Adipocytes, through the release of growth factors, exosomes, and metabolic symbiosis, promote tumor development. On the other hand, CSCs can reprogram the phenotype of neighboring adipocytes, characterized by an increase in lipolysis and overexpression of proinflammatory adipokines (such as growth factors and chemokines) and proteases [54] (Figure 1). These reactive adipocytes, known as CAAs, contribute to acquisition of the hallmark traits of cancer, such as increased angiogenesis, invasion, metastatic potential, and therapy resistance [55]. The presence of CAAs has been demonstrated in multiple cancers, including breast, ovarian, colon, prostate, and pancreatic tumors [53,56]. The scientific interest in these cell populations is justified by the fact that they could represent an alternative therapeutic target in the treatment of tumors.

CAAs can be distinguished from mature adipocytes according to their morphology and phenotype during the dedifferentiation process. The fibroblast-like morphology, typical of CAAs, is due to de-lipidation and alteration of the fat storage capacity, as found in patients with colorectal cancer [53]. In addition, compared to mature adipocytes, CAAs are smaller, have an increased number of mitochondria, show lower expression of differentiation markers, and exhibit a proinflammatory phenotype [3]. Among the adipokines secreted by CAAs, leptin, matrix metalloproteinase (MMP)-11, CCL2, chemokine ligand 5 (CCL5), IL-6, IL-1β, and TNF-α have been identified (Figure 1) [57,58,59].

These reactive adipocytes not only contribute to energy storage, thanks to the uptake of lipids, such as triacylglycerols (TAGs), which can be released in the microenvironment in the form of free fatty acids (FFAs), but also represent a source of a plethora of factors that induce both local and systemic effects [60].

CAAs adopt a dedifferentiation phenotype following tight crosstalk with CSCs. In particular, the expression of peroxisome proliferator-activated receptor γ (PPARγ) and CCAAT/enhancer-binding protein β (c/EBPα) is markedly inhibited, with a subsequent reduction in the mRNA levels of two markers of mature adipocytes, such as fatty-acid-binding protein 4 (FABP4) and hormone-sensitive lipase (HSL) [61,62]. The decrease in lipid droplets and cell size leads to the release of FFAs and ketone bodies, crucial factors for CSC metabolism in the promotion of tumor progression [59]. Another distinctive trait of CAAs is the increased production of ECM-related molecules, such as MMP-11, PAI-1, MMP1, fibroblast-activating protein (FAP), and collagen VI, which are involved in ECM remodeling, thus representing a crucial step in tumor progression [63]. In CAAs, the production and release of leptin and resistin is increased while the expression of adiponectin and adipokines with antitumorigenic functions is reduced [55].

### 4.1. Mechanisms and Regulation of CAA Activation

The acquisition of an activated phenotype by mature adipocytes is attributed to molecules that are released by tumor cells (Figure 1). However, the mechanisms underlying CAAs’ activation remain to be fully elucidated. Few studies have highlighted the role of molecules and pathways involved in the arrest of the adipogenesis process as a possible outcome of CAA development. In this context, the Wnt signaling pathway plays a key role as an inhibitor of adipogenesis and, therefore, has also been evaluated in adipocyte dedifferentiation. WNT3A [64] and WNT5A [65,66] have been identified as crucial players in the induction of an adipocyte dedifferentiation phenotype. Interestingly, the inhibition of Wnt signaling reduced the dedifferentiation rate and re-accumulation of lipids in fat cells [67]. Consistent with these results, exogenous administration of WNT3A was sufficient to restore the CAAs’ phenotype [66]. Other potential modulators released by CSCs that may trigger the reprogramming of mature adipocytes toward CAAs are represented by exosomes. Recent studies have shown that hepatocarcinoma cells (HepG2) release exosomes that, when internalized by adipocytes, can induce a proinflammatory cytokine expression profile similar to that of CAAs [68]. Some researchers suggested that IL-6, miRNA-144, and mir-126, which are released by breast cancer cells, are possible inducers of the CAA phenotype by reducing PPARγ expression and promoting adipocyte dedifferentiation [69]. Hu and colleagues showed that IL-6 contained within exosomes and released by Lewis lung carcinoma (LLC) cells activates the Jak/Stat3 pathway in adipocytes, inducing lipolysis [70].

Another two potential signaling pathways involved in adipocytes’ dedifferentiation are the Notch [71] and TGF-β pathways [72]. TGF- β is known to have a key role in adipose tissue remodeling via the activation of extracellular matrix molecules, such as collagen I and VI [73], and is probably involved in the activation of MMP11, a negative regulator of adipogenesis and inducer of dedifferentiation [74].

CD26, also known as DDP4 (dipeptidyl peptidase 4), is another molecule that participates in the induction of dedifferentiation of adipocytes. DDP4 is a transmembrane serine peptidase involved in ECM degradation by cleaving collagen, whose role in adipose tissue concerns adipose tissue remodeling and the regulation of cell plasticity by regulating C/EBP expression [75].

### 4.2. CAA-Secreted Molecules: A Key Role in Tumor Progression

In vitro and in vivo studies have shown the role of adipokines/cytokines, hormones, and metabolic substrates released by CAAs in influencing the hallmarks of cancer cells. Interestingly, a better characterization of the molecular events induced by CAAs, underlying the adipocyte–CSCs interplay, could elucidate the multistep process of tumorigenesis, paving the way for the development of new therapeutic strategies [60]. Importantly, CAAs exhibit dysregulated production of adipokines, both in terms of the quality and quantity of released factors, compared with mature adipocytes [57].

#### 4.2.1. Adipokines

Increased expression of leptin has been found in tumors compared to normal tissue, and its presence has been correlated with poor prognosis and a more invasive phenotype [76]. Leptin, a hormone encoded by the obese gene (OB), regulates the energy balance by binding to its receptor (OB-R), which is expressed in the hypothalamic arcuate and paraventricular nucleus [77]. Systemic levels of leptin are strictly dependent on the amount of body fat, with its secretion stimulated by insulin [78] and tumor necrosis factor-α (TNF-α), and inhibited by catecholamines [79]. Mechanistically, leptin binds its receptor OB-R in CSCs by promoting activation of the JAK/STAT3, c-Jun, and Akt pathways, thus inducing the transcription of IL-6, TGF-β, and MMP, which drive cancer cell proliferation and invasion. Consequently, in a mouse model, the absence of OB-R resulted in reduced tumor growth and invasiveness by reducing JAK/STAT3 and ERK1/2 signaling [80]. Another interesting role of leptin consists of modulation of the metabolic signature of CSCs. In breast cancer, Park et al. demonstrated that an absence of the leptin receptor led to a decrease in glycolysis and an increase in oxidative mitochondrial processes [80]. Moreover, Liu et al. highlighted that in melanoma cells, leptin can contribute to chemotherapy resistance by inhibiting apoptosis through activation of the Akt and MEK/ERK pathways. Leptin increased autophagic proteins in myeloma cells and decreased the apoptotic effect of conventional drugs, such as dexamethasone or doxorubicin, melphalan, and bortezomib, in tumor cells [81].

Adiponectin is another adipokine that is known to be a regulator of energy and nutrient homeostasis and an anticancer agent. In hepatocarcinoma, adiponectin can abolish the effects of leptin by inhibiting the proliferation and invasion of cancer cells [82]. Interestingly, the imbalance between leptin and adiponectin is a crucial factor in promoting tumor growth. In this regard, obese patients with elevated leptin levels and decreased adiponectin expression are more likely to develop tumors [83]. Patients with colorectal cancer have been found to express low levels of adiponectin, which correlates with a better prognosis. This study suggests that adiponectin can be considered as a promising biomarker of prognosis in colorectal cancer [84]. Conversely, a possible role of adiponectin in the promotion of cancer growth has been demonstrated [85]. The existence of two adiponectin forms may clarify these opposite roles. The complex and controversial role of leptin and adiponectin in supporting tumor growth needs further investigation to develop innovative therapeutic strategies [86].

#### 4.2.2. Cytokines

In addition to leptin and adiponectin, CAAs secrete different cytokines compared to mature adipocytes. Among them, we identified IL-6 as a proinflammatory cytokine that mediates the activation of Stat-3 in cancer cells, thus promoting tumorigenesis. In particular, the IL-6/Stat-3 axis has been described as the main mechanism through which adipocytes induce the EMT phenotype in tumor cells [87]. Interestingly, IL-6 together with leptin can increase the expression of procollagen-lysin-2-oxoglutarate 5 dioxygenase (PLOD2), a protein involved in ECM remodeling, and therefore has a decisive role in cancer cell invasion and EMT [88].

#### 4.2.3. Metabolites

According to the Warburg theory, cancer cells were thought to mainly depend on glycolytic metabolism, based on the conversion of glucose into lactate, which occurs independently from the presence of oxygen [89,90]. Recently, cancer cells, particularly CSCs, have been strongly linked to lipid metabolism [91,92]. In this scenario, CSCs have been shown to adopt different mechanisms to reprogram their metabolic pathways. Tumor cells require fuel energy to sustain biomass production and to promote migration, invasion, and metastasis. In this context, CSCs use metabolites released by stromal cells as substrates for anabolic metabolism. It is widely recognized that de novo lipogenesis, alteration in fatty acid storage, and β-oxidation are pivotal biological processes that support metabolic reprogramming, which is considered a typical hallmark of CSCs [60]. FFAs, released by CAAs in the surrounding microenvironment, are taken up by CSCs through several transporters, such as CD36, also called fatty acid translocase, FATPs (fatty acid transport proteins), and FABPs (fatty-acid-binding protein) [93]. Overexpression of CD36 has been found in several tumors, such as breast and ovarian cancers, and has been associated with poor prognosis [94,95]. The mechanisms by which FAs promote tumor progression are multiple: epigenetic changes, increased reactive oxygen species (ROS) expression, and metabolic remodeling [93]. Due to the significant role of FA in contributing to tumor pathogenesis, significant interest has been shown by the medical community in the development of therapeutic strategies that reprogram the metabolic fate of FAs. In this regard, inhibitors of enzymes involved in FA synthesis and the uptake of exogenous lipid have been developed.

Another metabolite that plays a key role in supporting cancer metabolism is glutamine, an amino acid that provides catabolic intermediates for the TCA cycle and has a key role in ATP mitochondrial generation in CSCs [96]. Several studies have demonstrated that glutamine promotes tumor growth and contributes to cancer therapy resistance. In pancreatic cancer cells, the secretion of glutamine by adipocytes was correlated with tumor proliferation while in leukemia cells, it was associated with a reduction in cytotoxicity induced by L-asparaginase [97,98].

Ketone bodies produce more ATP and consume less oxygen compared to glycolysis. Ketone bodies are catabolites produced by FA β-oxidation or aerobic glycolysis. In breast cancer, β-hydroxybutyrate, a ketone body released by adipocytes, has been shown to promote cancer proliferation and growth-inducing expression of its receptor, MCT2. Moreover, β-hydroxybutyrate favors the transcription of some genes, such as Il-1 β and lipocalin-2, through epigenetic regulation. High expression of MCT-2, Il-1 β, and lipocalin-2 is, indeed, correlated with poor prognosis in breast cancer patients [99].

#### 4.2.4. Exosomes

Emerging elements involved in adipocytes–cancer cells crosstalk include exosomes and miRNA. Exosomes are microvesicles with a diameter of 30–100 nm that contain mRNA, miRNA, lncRNA, and enzymes. In melanoma, these vesicles have been shown to transport proteins involved in the metabolic reprogramming of CSCs [100]. In a lung cancer model, exosomes carry MMP3 to CSCs, promoting tumor progression and metastasis in vivo through the activation of MMP9 [101]. Overall, these data highlight that TME is characterized by multiple scenarios of non-mutually exclusive events that arise from crosstalk between CAAs and CSCs. Despite the extrinsic and intrinsic factors elucidated in adipocyte dedifferentiation and CAAs’ involvement in tumor growth, progression, and drug resistance, further studies are needed to understand how CAAs can be targeted to develop an effective therapeutic strategy and identify a tumor cure.

## 5. Evolution of Adipose Tissue Microenvironment Heterogeneity

The architecture of the tumor-associated extracellular matrix is dramatically degenerated compared to normal tissue, with higher levels of tenascin, periostin, SPARC, and collagen, and different compositions of macromolecules, such as proteins, glycoproteins, proteoglycans, and polysaccharides. ECM is also rich in cytokines, chemokines, growth factors, and matrix remodeling enzymes produced by tumor-associated stromal cells [102,103], such as fibroblasts, endothelial cells, adipocytes, and infiltrating immune cells. Cancer cells recruit stromal cells and force them to form a protective and enriched TME, establishing dynamic and bidirectional tumor–stroma signaling [104,105,106].

Indeed, some of the most aggressive cancers have been shown to undergo lipid metabolism reprogramming (regulation of extracellular lipid uptake or de novo fatty acid synthesis pathways) to resist nutrient and oxidative stress [107]. Moreover, AT-MSCs and brown adipocytes have an outstanding ability to share their mitochondria (MT) with different target cells and thus influence their biological outcomes [108]. MT, also called the ‘powerhouse of the cell’, play a fundamental role in providing energy that is essential for all cellular functions [109]. Under pressure from TME, MT undergo complex reprogramming that includes changes in the organelle dynamics, metabolic activity, apoptosis control, and redox status of tumor cells [110,111].

Recently, it has been demonstrated that breast cancer cells stimulate lipolysis in adipocytes, leading to the release of FFAs, which are in turn taken up by tumor cells and promote the oxidation of β-fatty acids. This dynamic interplay was defined as complex metabolic symbiosis by Wang and colleagues [112]. Reprogramming of fatty acid metabolism in cancer cells promotes cancer progression and metastasis and contributes to the development of TME [93].

Tumor homing is a multistep process in which cancer cells spread from the primary tumor to a preferentially distant organ. The execution of this complex program requires the presence of a protective environment in the host tissue that allows cancer cells to colonize and grow. Accordingly, adipocytes contribute critically to the development of melanoma metastasis in bone marrow. Adipocytes are the most representative stromal cells in the bone marrow, and their number increases with age. In a comparison of older and younger patients, the incidence of melanoma bone metastasis was observed to increase with age, suggesting an indirect correlation with adipose tissue size [112]. In addition, adipose tissue in the periglandular regions and visceral omentum represents a preferential site for breast, prostate, and ovarian cancers [113].

During cancer progression, adipose tissue plays an important role, both influencing the phenotype of cancer cells and leading to TME instability that affects the activity of other stromal cells. CAAs and cancer-associated fibroblasts (CAFs) share some similarities in protumorigenic functions, suggesting their synergistic activity in promoting tumor progression, the metastatic process, and tumor resistance to standard treatments. Although several studies have indicated the role of CAAs and CAFs in TME, little is known about their direct interplay and mutual influence. Activated fibroblasts assume a multi-spindle shape, lose lipid droplets with their retinoid content, and express positivity for α-smooth muscle actin (α- SMA). The phenotypic changes observed in CAFs are associated with increased production of extracellular matrix components compared to normal fibroblasts [114].

A hallmark of pancreatic cancer is the abnormal deposition of fibrous tissue by CAF, termed the desmoplastic response. Increased collagen deposition leads to an alteration in the biochemical properties of ECM, making the tissue stiffer and reducing blood flow and permeability to chemotherapeutic agents [115,116]. In addition, tumor-associated ECM represents a reservoir of growth factors and proinflammatory cytokines that are simultaneously produced in large quantities by activated fibroblasts. Several findings indicate that in colorectal cancer patients, high expression of the CAF signature is associated with high levels of microenvironment-derived growth factors, such as HGF, FGF, VEGF, and EGF, which represent an important source of innate and acquired resistance to target therapies [117,118]. As a result of their tumor-promoting function, increased activity of CAFs and CAAs has been clinically associated with worse outcomes in cancer patients [119,120].

Dynamic and bidirectional tumor–stroma signaling also involves endothelial cells (ECs), since angiogenesis represents a critical step during tumor progression (Figure 2).

ECs in healthy tissue are usually quiescent and await activating inputs. The secretory-modulator factors present in TME stimulate reactivation of ECs, promoting their proliferation and migration, reducing the expression of critical fate markers, and inducing the acquisition of adhesion molecules that are attractive for leukocytes [121]. Consequently, tumor-associated ECs dramatically affect all stages of disease. The formation of new vessels, called tumor neo-angiogenesis, plays an important role in both the growth of the primary tumor and the metastatic dissemination of cancer cells [122]. Many of the key factors responsible for the activation of ECs are produced by CAAs and CAFs. 

CAAs, through the release of large amounts of vascular endothelial growth factor (VEGF), fibroblast growth factor-2 (FGF2), and adipokines, such as leptin, adiponectin, resistin, and IL-6, in TME induce a pro-angiogenic microenvironment that is responsible for the formation of new blood vessels [123] (Figure 2). Tumor vessels are also termed ‘mosaic vessels’ due to their atypical organization and morphology, in which cancer cells are exposed to the lumen of the tumor blood vessels due to an incomplete endothelial lining [124,125]. Thus, this altered vasculature mainly has dual implications in cancer progression, hindering uniform drug delivery and promoting the escape of cancer cells from the primary tumor [126,127]. 

Finally, the inflammatory state of TME is also influenced by the activity of CAAs. As previously described, CAAs modulate the innate and adaptive antitumor immune response through the release of a large number of inflammatory cytokines, including IL-6, IL-8, IL1-β, and TNF-α [128] (Figure 2). One of the components of the innate immune system is represented by neutrophil granulocytes, the concentration of which increases in the blood of patients during the progression of cancer [129]. CAAs are indirectly linked to neutrophils through the release of chemokines and FFAs, which maintain immune cells’ activated state and increase glycolytic flux. In pancreatic cancer, CAAs-derived IL-1β recruits neutrophil granulocytes, which in turn induces the activation of pancreatic stellate cells, promoting a desmoplastic response and disease progression [130]. Another example of the link between CAAs and the innate immune system was described in the study by Liang et al., in which CAAs-derived IL-8 increased neutrophil recruitment to tumor tissue. Moreover, chemokines induced the expression of integrin LFA-1 on the surface of immune cells, which mediated their adhesion to endothelial cells and promoted the extravasation of melanoma cancer cells [131]. FFAs released by CAAs also impair the functions of natural killer cells (NKs) by reprogramming their metabolism [132]. In obese patients, NKs accumulate in visceral adipose tissue and release increased IFN-γ under the pressure of stress conditions, which promotes macrophage polarization toward an M2 phenotype [131,133]. Mononuclear phagocytes play a critical role in maintaining tissue homeostasis by activating various functional programs in response to external signals. Differentiated monocytes are mainly classified into M1 macrophages, which secrete proinflammatory cytokines, such as TNF-α and IL-6, and M2 macrophages, which mediate an anti-inflammatory effect via IL-10 [134]. Both TME and tumor cells drive monocytes to acquire an M2 phenotype. CAAs influence macrophage polarization through the release of FFAs imported into cells by the scavenger receptor CD36. The uptake of triacylglycerols by macrophages induces mitochondrial biogenesis and promotes the switch to the M2 phenotype [135] (Figure 2). Metabolic reprogramming also dampens T cell function, which affects the antitumor immune response. Effector T cells are responsible for executing and coordinating the activity of the immune system. Since activated effector T cells use the process of glycolysis to carry out their effector function, metabolic reprogramming in the presence of TME T cells impairs immune cell development and functionality [136,137]. CD8+ T cells’ survival and functionality is influenced by cell metabolism. In normal conditions, naive CD8+ T cells are characterized by low metabolism and are dependent on mitochondrial oxidative phosphorylation as their main energy source. Elevated levels of FFAs and lactate in TME lead to immune suppression by preventing the activation of cytotoxic T lymphocytes and dendritic cells [138]. Moreover, PD-L1 expression in CAAs impairs CD4+-positive T cells by increasing glycolysis and fatty acid oxidation in several solid tumors. Interfering in the PD-1/PD-L1 pathway may indirectly affect T cell metabolism and function [139]. CAAs also indirectly act on Tregs by increasing their activity as sentinels of the immune response to prevent autoimmune diseases. The differentiation and survival of Tregs is dramatically affected by the degradation of FFAs. In breast cancer, an increased Treg population was observed in TME in comparison with normal tissue. Moreover, higher expression of FFA-binding proteins in immune cells was identified compared to the expression in Tregs derived from peripheral blood [139]. Moreover, increased aerobic activity persists in CAAs, generating a lactate-rich environment that promotes the immunosuppressive response of Tregs [140]. However, whether lactate and FA metabolism play a role in modulating the antitumor response of the immune system requires further investigation.

## 6. Targeting of Adipose Tissue–Tumor Crosstalk Mediators

Of note, some therapy resistance mechanisms exhibited by tumor cells are strictly linked to the metabolic switch from glycolytic to lipid metabolism, thus underlying the clinical impact of enzymes that regulate the metabolic pathways [141,142,143,144]. These enzymes are currently being studied as a possible target of novel antitumor therapeutic approaches, as a single treatment or as a chemotherapy adjuvant for therapy-resistant patients [145,146,147,148].

### 6.1. FASN-Targeting Drugs

In antitumor therapeutic strategy, fatty acid synthase (FASN) has received extensive interest. First-generation FASN-targeting drugs, such as cerulenin, C75, and Orlistat, showed promising results in preclinical studies, with a reduction in tumor xenograft growth. However, the use of FASN-targeting drugs in clinical trials has been challenging, with the onset of different side effects, such as anorexia and weight loss [149].

Recently, a new generation of FASN inhibitors, such as TVB-3166 and TVB-2640, have displayed an effective antitumoral potential in preclinical colorectal and breast cancer models, and satisfactory tolerability and limited side effects in clinical trials [150].

Interestingly, in the light of the correlation between FASN expression and signaling downstream of HER-2, it could be useful to develop FANS-targeting therapies to better stratify patients based on their possible response to standard chemotherapy. Nowadays, a clinical trial in the phase 2 stage assessing the combinatorial effects of TVB-240 and chemotherapy in HER-2-positive breast cancers is currently ongoing (NCT 3179904).

In the context of chemoresistance, an important aspect that should be taken into consideration concerns the ability of adipocytes to metabolize chemotherapeutic agents, thus decreasing their bioavailability and paving the way for the onset of chemoresistance phenomena [151]. In this regard, combination treatment with FAS enzyme inhibition associated with standard chemotherapy for the treatment of ovarian cancer resulted in a successful therapy approach both in in vitro and in vivo settings for therapy-resistant ovarian cancer cells [152,153].

### 6.2. Targeting of CD36

CAAs exert immunomodulatory activity during cancer progression and represent a promising therapeutic target to improve immunotherapy. Interfering with CAAs may indirectly affect T cell metabolism and function. Two different strategies have been investigated to target adipocytes in TME: specifically interfering with adipocytes or blocking CAAs-derived signals. However, normalization of CAA activity is difficult due to the loss of adipocyte-specific markers and the potential of damaging healthy organs because of the ubiquitous nature of adipose tissue [111].

Indirect targeting of CAAs represents the more explored strategy. In breast cancer, for example, inhibition of CD36 fatty acid transport proteins is associated with a decrease in intratumoral Tregs and an increase in antitumoral infiltrating effector T cells. Moreover, Wang et al. combined the antitumor effect of CD36 monoclonal antibody with anti-PD-1 therapy. This dual targeting abrogated the reprogramming of TME toward an immunostimulatory state and enhanced the antitumor activity mediated by the single immunotherapy drug [154]. CD36 is a transmembrane glycoprotein, belonging to the class B scavenger receptor family, implicated in FA uptake and angiogenesis, cell adhesion, and antigen presentation. Ovarian cancer cells expressed high CD36 levels when co-cultured with primary human omental-derived adipocytes, resulting in increased FA uptake. Interestingly, in the presence of CD36 inhibitors and short hairpin RNA knockdown, cancer cells did not acquire a malignant phenotype. Moreover, in an in vivo model, blocking of CD36 led to a reduction in tumor growth [94]. This study suggests that CD36 inhibition could represent a valid therapeutic tool for limiting tumor aggressiveness.

### 6.3. PPAR-γ Antagonists

The formation of mature adipocytes is finely regulated by several adipogenic transcription factors, such as PPARγ, CCAAT/enhancer-binding proteins (C/EBPs), Krüppel-like factors (KLFs), and sterol regulatory element-binding protein 1c (SREBP1c), which act in concert to drive MSC differentiation [21,22]. PPARγ is a regulator of lipid and glucose homeostasis and can upregulate tumor suppressor genes, such as BRCA1 and PTEN. In this regard, PPARγ antagonists have been suggested as possible drugs to target the crosstalk between adipocytes and CSCs. GW9662 is an example of a PPARγ antagonist, which is also known as a suicide inhibitor. When administered to FVB mice with induced mammary tumors, GW9662 sensitized ER-responsive tumors to fulvestrant therapy [155]. Cheng and co-workers demonstrated that GW9662 can inhibit PPARγ function, thus significantly impairing bladder cell proliferation in vitro, and decreasing tumor growth in vivo [156].

### 6.4. Targeting of miRNA

Exosomes, for the delivery of miRNA, represent a new promising therapeutic target strategy for tumor treatment. Indeed, miRNAs are very stable and can regulate cell proliferation and differentiation and induce chemosensitivity in cancer cells. In hepatocarcinoma, the downregulation of liver-specific miRNA (miRNA-122) is related to poor prognosis and invasiveness. A study carried out by Lou and co-workers demonstrated that miRNA-122-enriched exosomes, released by adipocytes, can sensitize the hepatocarcinoma to treatment with sorafenib [157]. In ovarian cancer cells, next-generation sequencing analysis showed high miRNA-121 expression in exosomes released by CAAs in the surrounding ovarian cancer cells’ microenvironment. In this study, the researchers conferred a decisive role in inducing chemoresistance to paclitaxel to miRNA-121 via downregulation of APAF1, a direct target of miRNA-121. These data suggest a possible strategy for reducing the chemoresistance induced by miRNA-121 in ovarian cancer cells through direct inhibition of miRNA-121 or by upregulation of APAF-1 [158].

### 6.5. Targeting of Metabolites

Among the metabolites released in TME, lactate may represent another therapeutic target for inhibiting immunosuppression mediated by CAAs. MTC1 is a transporter for monocarboxylates, such as lactate, whose expression is increased in tumors compared to normal tissues. AZD3965 is a potent and selective inhibitor of MTC1 that blocks the movement of chemical compounds in and out of cells, killing cancer cells. Polanski et al. demonstrated the therapeutic potential of this critical drug in non-small cell lung cancer (NSCLC) [159]. A clinical trial is currently underway to test the antitumor efficacy of the MTC1 inhibitor in advanced solid tumors (NCT01791595). Alternatively, regression of NSCLC tumors and activation of the immune system have been observed through inhibition of lactate dehydrogenase, which is involved in the conversion of pyruvate to lactate [160].

Collectively, it is important to underline that adipocytes and CAAs and their adipokines and metabolites represent promising candidates for targeting therapeutic strategy in cancer treatment.

## 7. Conclusions

Although several mediators released by cancer-associated adipocytes (CAAs) are considered promising therapeutic targets for anticancer therapy, there is an urgent need to further investigate the molecular mechanisms underlying the dynamic interplay between CSCs and adipose cells to propose innovative strategies based on the inhibition of specific signaling pathways or on nutritional interventions.

## Figures and Tables

**Figure 1 biomolecules-12-00702-f001:**
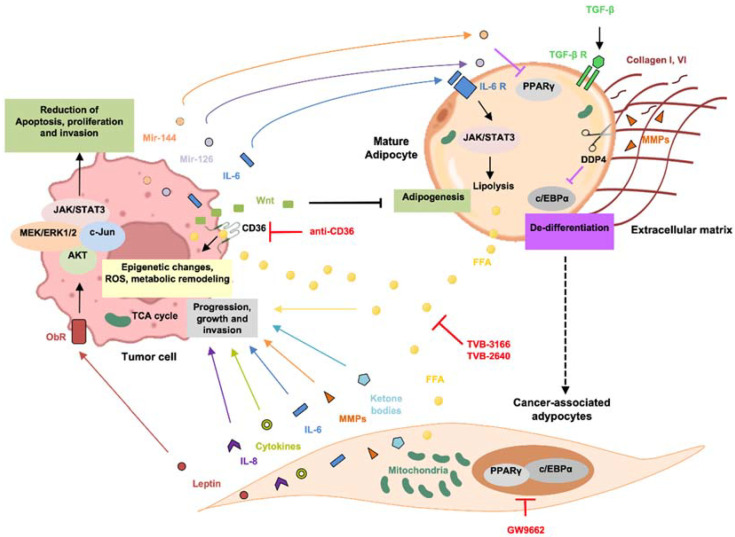
Schematic diagram showing the direct crosstalk between tumor cells, mature adipocytes, the cancer-associated adipocyte (CAA) phenotype, and possible targeting therapeutic strategies. Drugs targeting the biological mediators of adipose tissue–tumor bidirectional crosstalk are shown in red.

**Figure 2 biomolecules-12-00702-f002:**
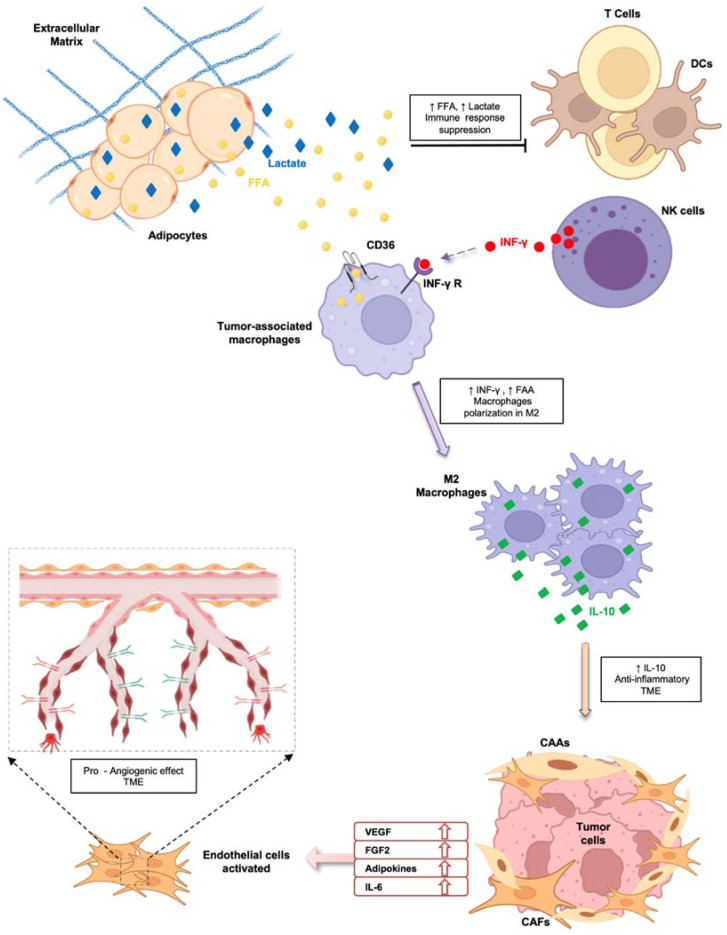
Schematic representation of indirect effects mediated by adipocytes and cancer-associated adipocytes on immune system cells, tumor cells, and endothelial cells, leading to cancer progression.

## Data Availability

Not applicable.

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
