# Peer review of "Targeting of the Peritumoral Adipose Tissue Microenvironment as an Innovative Antitumor Therapeutic Strategy"

_biomolecules, 2022, doi:10.3390/biom12050702_

Round 1

Reviewer 1 Report

The authors describe a timely subject, the relation between obesity and cancer, which is a field that requires ample attention with the increasing world-wide obesity pandemic. The authors provide different views and describe the broader context of adipocytes, CAA, CAF, TME in relation to tumorgenesis. 

However, the manuscript requires major revisions before publication can be considered:

  • Major revision of the English language, particularly section 4 is hardly readable
  • Particularly in section 4, statements are made without proper referencing, this should be adjusted. In addition, the entire manuscript should be checked regarding proper referencing of statements.
  • The figures are rather complex, advice is to simplify these to focus on the key principles that are discussed.
  • Several times, redundant information is provided on similar concepts. Advice is to condens the information in a straightforward manner and omit the scattered manner in the current manuscript.

Author Response

  • The authors describe a timely subject, the relation between obesity and cancer, which is a field that requires ample attention with the increasing world-wide obesity pandemic. The authors provide different views and describe the broader context of adipocytes, CAA, CAF, TME in relation to tumorgenesis. 

    However, the manuscript requires major revisions before publication can be considered:

    Major revision of the English language, particularly section 4 is hardly readable

We thank the reviewer for the constructive comments. According to the reviewer’s suggestion, the manuscript has been fully revised, with particular focus on section 4. Would you need the original file with the track changes we will be more than happy to provide it, in order to highlight all the adjustments made though the manuscript.

  • Particularly in section 4, statements are made without proper referencing, this should be adjusted. In addition, the entire manuscript should be checked regarding proper referencing of statements.

According to this reviewer's comment, the entire manuscript has been checked regarding proper referencing of the reported statements, with addition of new references and the replacement of those considered usuitable.

  • The figures are rather complex, advice is to simplify these to focus on the key principles that are discussed.

We appreciate the comment of this reviewer. After having carefully checked the figures, in particular Figure 1, which is more complex in terms of described mechanisms, we decided to remove some graphic elements to make it more clear and less confusing. Although we consider all the molecular mechanisms described in the figures crucial for the readers to have a complete overview of the "complex" cross-talk between cells within TME, we are ready to apply further modifications should the Editor deem it necessary.

  • Several times, redundant information is provided on similar concepts. Advice is to condens the information in a straightforward manner and omit the scattered manner in the current manuscript.

We agree with this reviewer regarding the presence of redundant information in the first version of submitted manuscript. According to this reviewer's comment, we have checked the whole manuscript, and removed all the redundant concepts (i.e., the findings previously reported in lanes 158-160 have been moved to lanes 68-73, while those described in lanes 218-227 were moved to lanes 206-213).

Moreover, according to Reviewer #2's suggestion, we have modified the structure of the manuscript, by introducing subparagraphs, which should make the manuscript clearer to the readers.

Reviewer 2 Report

This study comprehensively describes the contribution of the relationship between CAA and CSCs to TME, and these possibility as therapeutic targets. The manuscript is well-written and provides the important findings. The reviewer has no concern and gives the following advice to make it easier to understand.

The section 4 “Cancer-associated adipocytes (CAAs): origin…” may be clearer if classified into the factors, such as “signals” and “metabolites” …etc.

Likewise, the section 6 “Targeting of adipose tissue-tumor crosstalk mediators” may also be clearer if classified by target molecules.

Author Response

  • This study comprehensively describes the contribution of the relationship between CAA and CSCs to TME, and these possibility as therapeutic targets. The manuscript is well-written and provides the important findings. The reviewer has no concern and gives the following advice to make it easier to understand.

We thank the reviewer for the appreciation of our review manuscript both in terms of quality and topics here described.

  • The section 4 “Cancer-associated adipocytes (CAAs): origin…” may be clearer if classified into the factors, such as “signals” and “metabolites” …etc.

According to the reviewer's suggestion, we have further divided the section 4 of manuscript into paragraphs and subparagraphs, which should make the manuscript clearer to the readers, in terms of the topics and findings here described. The manuscript now contains the following structure:

4. Cancer-associated adipocytes (CAAs): origin and role in cancer progression; 4.1 Mechanisms and regulation of CAAs’ activation; 4.2 CAA-secreted molecules: a key role in the tumor progression; 4.2.1 Adipokines; 4.2.2 Cytokines; 4.2.3 Metabolites; 4.2.4 Exosomes; 

  • Likewise, the section 6 “Targeting of adipose tissue-tumor crosstalk mediators” may also be clearer if classified by target molecules.

We agree with this reviewer that the addition of subparagraphs even in section 6 could help the readers in better appreciating the specific results here presented. Thus, we have divided section 6 as follow:

6. Targeting of adipose tissue-tumor crosstalk mediators; 6.1 FASN-targeting drugs; 6.2 Targeting of CD36; 6.3 PPAR-γ antagonists; 6.4 Targeting of miRNA; 6.5 Targeting of metabolites.

Round 2

Reviewer 1 Report

Comments on the first manuscript have been taken into account. This has increased the quality of the manuscript.